# Decreased Vitamin D Levels and Altered Placental Vitamin D Gene Expression at High Altitude: Role of Genetic Ancestry

**DOI:** 10.3390/ijms24043389

**Published:** 2023-02-08

**Authors:** Eugenia Mata-Greenwood, Hans C. A. Westenburg, Stacy Zamudio, Nicholas P. Illsley, Lubo Zhang

**Affiliations:** 1Lawrence D. Longo Center for Perinatal Biology, School of Medicine, Loma Linda University, Loma Linda, CA 92350, USA; 2Placental Research Group LLC, Maplewood, NJ 07040, USA

**Keywords:** high altitude, genetic ancestry, vitamin D, metabolism, placenta

## Abstract

High-altitude hypoxia challenges reproduction; particularly in non-native populations. Although high-altitude residence is associated with vitamin D deficiency, the homeostasis and metabolism of vitamin D in natives and migrants remain unknown. We report that high altitude (3600 m residence) negatively impacted vitamin D levels, with the high-altitude Andeans having the lowest 25-OH-D levels and the high-altitude Europeans having the lowest 1α,25-(OH)_2_-D levels. There was a significant interaction of genetic ancestry with altitude in the ratio of 1α,25-(OH)_2_-D to 25-OH-D; with the ratio being significantly lower in Europeans compared to Andeans living at high altitude. Placental gene expression accounted for as much as 50% of circulating vitamin D levels, with *CYP2R1* (25-hydroxylase), *CYP27B1* (1α-hydroxylase), *CYP24A1* (24-hydroxylase), and *LRP2* (megalin) as the major determinants of vitamin D levels. High-altitude residents had a greater correlation between circulating vitamin D levels and placental gene expression than low-altitude residents. Placental 7-dehydrocholesterol reductase and vitamin D receptor were upregulated at high altitude in both genetic-ancestry groups, while megalin and 24-hydroxylase were upregulated only in Europeans. Given that vitamin D deficiency and decreased 1α,25-(OH)_2_-D to 25-OH-D ratios are associated with pregnancy complications, our data support a role for high-altitude-induced vitamin D dysregulation impacting reproductive outcomes, particularly in migrants.

## 1. Introduction

Residents at high altitude are at higher risk of reproductive challenges including intrauterine growth restriction, stillbirth, preeclampsia (de novo maternal high blood pressure and organ system damage), and perinatal mortality and morbidity [1,2,3,4,5]. Multiple studies have shown multigenerational high-altitude residents, such as Tibetans and Andeans, compared to newcomer residents, such as Han and Hispanics, are less affected by high-altitude chronic hypoxia-induced effects in pregnancy [6,7]. Although physiological adaptations explain some of the reproductive differences between ethnic groups to high-altitude residence [6,8], the molecular and genetic mechanisms remain elusive [9].

Vitamin D has important roles in maternal-fetal tolerance and placental development [10,11]. Vitamin D deficiency has been associated with intrauterine growth restriction, recurrent miscarriages, gestational diabetes, and pre-eclampsia [12,13,14,15]. Furthermore, maternal vitamin D deficiency has been also linked to postnatal infectious diseases, autoimmune diseases (diabetes type I), asthma, and obesity in offspring [16,17,18]. On a global basis, vitamin D deficiency during pregnancy is highly prevalent [19,20,21]. Several developmental (age, pregnancy), lifestyle (diet, exercise, sun exposure), and environmental factors (latitude, altitude, climate, season) are known to affect vitamin D status [22,23]. Most human studies have observed an important correlation between sun exposure and vitamin D levels accounting for more than 50% of changes in vitamin D status [22,23,24]. Ultraviolet B-light exposure is increased at higher altitude; however, several studies have shown significant vitamin D deficiency in humans, including pregnant women, living in high-altitude locations [25,26,27,28,29,30]. This renders Vitamin D deficiency a potentially important mediator of adverse pregnancy outcomes associated with high-altitude residence.

Vitamin D metabolism is greatly altered during mammalian pregnancy [31,32,33]. In humans and other mammal species, the maternal plasma levels of calcium and the abundant precursor 25-(OH)-D remain at pre-pregnancy levels while the potent 1α,25-(OH)_2_-D metabolite levels increase 2–3-fold compared to pre-pregnancy levels as early as in the first trimester [31,33]. The placenta is known to express nearly all vitamin D-related genes and to actively participate in vitamin D homeostasis during pregnancy [32,33]. Vitamin D_3_ (cholecalciferol) can be endogenously produced from 7-dehydrocholesterol by sun exposure in the maternal skin epidermis or obtained from dietary sources [34,35]. 7-Dehydrocholesterol reductase (DHCR7) can sequester this metabolite to synthesize cholesterol. Vitamin D is endogenously activated by 25-hydroxylase (CYP2R1) producing 25-OH-D, followed by a second hydroxylation by 1α-hydroxylase (CYP27B1) that yields the most active metabolite of vitamin D: 1α,25-(OH)_2_-D [34,35]. In the blood, vitamin D metabolites are transported bound to albumin and the vitamin D binding protein (VDBP/GC). Megalin (LRP2) and cubulin (CUBN) bind to the VDBP/25-OH-D complex acting as efficient transporters. Inside the nucleus, 1α,25-(OH)_2_-D binds and activates the vitamin D receptor (VDR) that acts as a transcription factor leading to gene expression changes. A highly sensitive gene upregulated by VDR is the vitamin D-inactivating 24-hydroxylase (CYP24A1), thereby providing a unique negative feedback control mechanism to prevent vitamin D toxicity [34,36]. The aim of the present study is to identify the effect of altitude and genetic ancestry on maternal/fetal vitamin D levels and placental vitamin D metabolism. We hypothesize that high-altitude residence will be associated with vitamin D deficiency, based on prior reports [22,23], and that Andean-ancestry women would demonstrate improved vitamin D homeostasis compared to women of European ancestry.

## 2. Results

### 2.1. Demographic and Clinical Characteristics

Our cross-sectional mother/neonate dyad study included four groups (*n* = 10/group): Andean ancestry at 400 m altitude (An-Low), Andean ancestry at 3600 m altitude (An-High), European ancestry at 400 m altitude (Eu-Low) and European ancestry at 3600 m altitude (Eu-High). The Eu-Low group was significantly younger than the Eu-High and An-High groups, otherwise, the groups had similar maternal age, pre-pregnancy BMI, and neonatal sex distributions (Table 1). There were multiple significant differences in fetal/neonatal outcomes. The Eu-High group delivered at a significantly earlier gestational age than the An-High and An-Low (Table 1), and the Eu-Low group also delivered at a significantly earlier gestational age than the An-Low (Table 1). The Eu-High group had a significantly lower birthweight percentile than the Eu-Low group (31 ± 19.7 vs. 63.6 ± 27.3, Table 1) and significantly lower birthweight than the other three groups (Table 1), although none of the 40 newborns were clinically considered growth restricted. Placental efficiency was estimated indirectly by the birthweight-to-placental weight ratio (BW:PW), with lower ratios indicating lower efficiency, i.e., more placenta is required to support a given fetal size. The BW:PW ratio did not differ between Andean and European groups, but was significantly decreased at high altitude in both ancestry groups (Table 1). The Eu-High group also had a significantly lower ponderal index than the Eu-Low and the An-High group (Table 1). Finally, although there were no differences in head circumference, the Eu-High group of newborns had a significantly lower abdominal circumference than the An-High and An-Low group (Table 1). Two-Way ANOVA suggested that altitude decreased placental efficiency, birthweight, and birthweight percentile, while genetic ancestry had a significant effect on birthweight and abdominal circumference. Ponderal index, birthweight, and birthweight percentile showed a significant interaction between altitude and ancestry.

### 2.2. Vitamin D Status in Bolivian Pregnancies

Near-term pregnancies at high altitude were characterized by significantly lower levels of vitamin D metabolites in mothers and neonates (Figure 1). The altitude-associated reduction in 25-OH-D was significant only in An-High mothers and their fetuses (Figure 1A,B). As a result, maternal and fetal/neonatal 25-OH-D plasma levels were significantly lower in the An-High group compared to the other three groups (Figure 1A,B). In contrast, maternal 1α,25-(OH)_2_-D levels were significantly lower in the Eu-High compared to the Eu-Low while the reduction in this metabolite between An-High and An-Low was not significant (Figure 1C). There were no differences in fetal/neonatal 1α,25-(OH)_2_-D levels (Figure 1D), or maternal VDBP levels (Figure 1G). Of note, vitamin D deficiency, defined as circulating levels of 25-OH-D less than 25 nM, was present in four mothers and five neonates of the An-High group, one mother and her neonate in the Eu-High group, one neonate in the An-low group and zero subjects of the Eu-low group. The effect of altitude across the four groups is better shown in the altered ratio of the two metabolites; 1α,25-(OH)_2_-D:25-OH-D. This ratio was similar at low altitude in the two ancestry groups, but it was significantly higher in An-High compared to Eu-High in both mothers (Figure 1E) and neonates (Figure 1F). Analysis of the effect of altitude and ancestry by two-way ANOVA revealed that only altitude had a significant effect on vitamin D levels (*p* < 0.05). However, it was ethnicity and its interaction with altitude that significantly influenced the ratio of the two metabolites (*p* < 0.05). Multivariable linear regression confirmed that altitude was the most significant determinant of maternal and fetal/neonatal 25-OH-D and 1α,25-(OH)_2_-D levels (Table 2). In addition, placental efficiency (BW:PW) was a second independent factor associated with fetal/neonatal 25-OH-D as well as maternal and fetal/neonatal 1α,25-(OH)_2_-D (Table 2).

Linear regression analysis showed a significant correlation between maternal and fetal/neonatal blood levels of 25-OH-D in all four groups (*p* < 0.001, Appendix AA–D). In contrast, the correlation between maternal and fetal/neonatal blood levels of 1α,25-(OH)_2_-D was only significant in the An-High group (r = 0.8519, *p* < 0.001, Appendix AF). Furthermore, there was no significant correlation between 25-OH-D and 1α,25-(OH)_2_-D levels, except in maternal and fetal/neonatal samples of the An-High group (Appendix AB,F) and fetal-neonatal samples of An-Low and Eu-Low (Appendix AE,G).

### 2.3. Placental Expression of Vitamin D-Related Genes

Altitude had a significant effect on the placental expression of vitamin D metabolic enzymes (Figure 2). The cholesterol-synthesizing enzyme *DHCR7* mRNA levels were significantly higher in An-High and Eu-High than in their respective low-altitude groups (Figure 2A). There were no significant differences among the four groups in the expression of *CYP2R1* and *CYP27B1* that synthesize 25-OH-D and 1α,25-(OH)_2_-D, respectively (Figure 2B,E). The vitamin D metabolite transporter *LRP2* mRNA was significantly higher in Eu-High than Eu-Low samples (Figure 2C), but there were no significant differences in *CUBN* transporter expression levels among the four groups (Figure 2D). The vitamin D inactivating enzyme *CYP24A1* expression was significantly higher in the Eu-High than the An-High group (Figure 2F). Placental *VDR* mRNA levels were also significantly higher in the high-altitude groups than in the corresponding low-altitude groups (Figure 2G).

Multivariable linear regression analyses did not yield a significant model of placental gene determinants of vitamin D levels for all 40 subjects. However, stratification by altitude uncovered significant placental vitamin D-relevant genes as determinants of vitamin D levels (Table 3). For low-altitude subjects, 26.5% of the maternal 25-OH-D variation could be explained by placental *CYP27B1* mRNA levels, while 22% of the fetal/neonatal 25-OH-D variation could be explained by placental *CYP2R1* mRNA levels (Table 3). Stronger correlations were found in the high-altitude group. Both maternal and fetal/neonatal 25-OH-D levels were significantly determined by placental *CYP24A1* and *CYP2R1* expression by 50.5% (maternal) and 35% (fetal/neonatal) (Table 3). Furthermore, 29.7% of maternal 1α,25-(OH)_2_-D levels were determined by placental *LRP2* and *CYP24A1* expression while there was no significant model of placental gene expression as determinants of fetal/neonatal 1α,25-(OH)_2_-D levels (Table 3). The correlation between vitamin D levels and individual placental gene expression (Figure 3) illustrates the differential gene determinants of vitamin D levels according to altitude shown in Table 3. First, placental *CYP2R1* and *CYP27B1* were differentially regulated by altitude: at low altitude, there was a significant negative correlation with maternal 25-OH-D levels (Figure 3A,C) while at high altitude, there was a positive correlation, although it was only significant for *CYP2R1* (Figure 3B) not *CYP27B1* (Figure 3D). In addition, there was a significant positive correlation between maternal/fetal 25-OH-D and 1α,25-(OH)_2_-D levels with placental *CYP24A1* expression in high-altitude samples but no correlation in low-altitude samples (shown for maternal 25-OH-D in Figure 3E,F). Finally, high-altitude, but not low-altitude, samples showed a significant positive correlation between maternal 1α,25-(OH)_2_-D levels and placental *LRP2* mRNA levels (Figure 3G,H).

## 3. Discussion

The present study has provided novel evidence of associations and potential mechanisms of high-altitude effects on maternal and fetal/neonatal vitamin D status and placental vitamin D metabolism. We confirmed our hypothesis that high-altitude residence is associated with decreased vitamin D levels in both mother and fetus with Andean-ancestry subjects having decreased 25-OH-D and European-ancestry subjects having decreased 1α,25-(OH)_2_-D. Our data support prior reports showing decreased 25-OH-D levels in indigenous South American populations living at high compared to low altitude [28,29,30]. Other studies report decreased 25-OH-D levels in Tibetans at high altitude and Turkish pregnant women at moderate altitude, but without comparisons of the same population at lower altitudes [25,26,27]. A study of European alpinists showed a significant decrease in 25-OH-D levels after a two-week climb [37]. It is thus consistent that both our multiple linear regression and two-way ANOVA identified high altitude as the most relevant factor affecting term pregnancy vitamin D levels. We identified placental efficiency as a secondary factor determining fetal 25-OH-D, and maternal/fetal 1α,25-(OH)_2_-D levels, which led us to evaluate placental gene expression of the vitamin D system. These data point to a pivotal role of the placenta in regulating the circulating levels of vitamin D metabolites in both the mother and fetus. Placental impacts on vitamin D status during pregnancy are reflected in the sudden drop in 1α,25-(OH)_2_-D levels shortly after delivery, and the correlation of placental gene expression with maternal/fetal vitamin D levels [31,32,33,38]. Placental efficiency was significantly decreased by high altitude in both genetic ancestry groups (Table 1). It is important to note that the original study revealed that for any given placental weight, Andean neonates at high altitude were >200 g heavier than their European counterparts, indicating a superior placental adaptation to high altitude in the Andean population [39]. Based on the beneficial effects of both 25-OH-D and 1α,25-(OH)_2_-D in various aspects of mammalian pregnancy [12,13], decreases in vitamin D status are likely an intermediary of the negative effects of high altitude in human pregnancy outcomes.

Of interest, we found that placental gene expression had a stronger correlation with maternal/fetal vitamin D levels at high compared to low altitude. Previous studies have shown positive correlations between maternal 25-OH-D levels and placental *LRP2*, *CUBN*, *CYP27B1*, and *CYP24A1* expression [38,40]. However, we did not find similar correlations in the low-altitude subjects, where, in fact, we observed negative correlations of maternal and fetal 25-OH-D levels with the expression of main metabolic genes *CYP2R1* and *CYP27B1*. These are novel findings that we believe could be due to the higher levels of both vitamin D metabolites present in this study’s low-altitude residents, which lead to compensatory downregulation of the two main synthesizing enzymes. The Bolivian low-altitude residing mothers had significantly higher 25-OH-D by 23% and 1α,25-(OH)_2_-D levels by 41% than in our California cohort of healthy mothers [41]. Furthermore, other than *CYP2R1* and *CYP27B1*, there were no other placental genes that correlated with vitamin D levels in low-altitude pregnancies. In contrast, mothers living at high altitude showed strong correlations between vitamin D metabolite levels and placental vitamin D-related gene expression, with the exception of *CYP27B1* (in contrast to low altitude). Multivariable linear regression confirmed that, at high altitude, *CYP24A1* and *CYP2R1* were strong determinants of maternal and fetal 25-OH-D levels, while *CYP24A1* and *LRP2* were determinants of maternal 1α,25-(OH)_2_-D. We hypothesize that mothers exposed to high-altitude environmental stressors (hypoxia, cold) experience a more stringent regulation of vitamin D status by the placenta.

The decreased vitamin D levels at high altitude can be partially explained by the changes in placental vitamin D-related gene expression. High-altitude residence was associated with the upregulation of placental *DHCR7* and *VDR* mRNA levels in both Andean- and European-ancestry groups. Placental *LRP2*, while elevated by altitude in both ancestry groups, was significantly increased only for the Eu-High group. Increased placental *DHCR7* expression could partially explain decreased 25-OH-D, since it sequesters the main precursor of vitamin D metabolites, 7-dehydrocholesterol, for cholesterol synthesis. Indeed, it has been reported that high altitude increases total serum cholesterol and other lipids [42,43]. LRP2 is an important transporter of nutrients and lipids, including vitamin D and cholesterol, which is essential for fetal growth [44,45]. Thereby, *LRP2* upregulation at high altitude is a beneficial response; however, this is the first study to show upregulation of this gene in high-altitude placentas. In silico analyses of *DHCR7* and *LRP2* identified multiple (>6) hypoxia-inducible factor alpha (HIF1α) consensus DNA response sites (A/GCGTG) in their gene promoters. In contrast, promoters of *CYP24A1*, *VDR*, *CUBN*, and *CYP2R1* had one or two HIF1α sites and *CYP27B1* had none. Therefore, it is possible that high altitude upregulates *DHCR7* and *LRP2* expression via this transcription factor, although HIF1α chromatin immunoprecipitation studies would be required to confirm this. In support of this hypothesis, we have previously shown that the placentas of high-altitude pregnancies show increased levels of HIF1α, its downstream gene targets, and correlations with birth outcomes [46]. Another interesting finding was the upregulation of placental *VDR* at high altitude in both genetic ancestry groups. Multiple studies on placental vitamin D metabolism in pregnancy complications demonstrated reduced placental *VDR* expression, in conjunction with vitamin D deficiency, in preeclampsia, intrauterine growth restriction, and preterm birth [47]. Therefore, high-altitude-induced upregulation of placental *VDR* could be a beneficial compensatory effect against decreased vitamin D status. This finding distinguishes altitude-associated growth restriction from pathological IUGR mechanisms [47].

We also uncovered an interaction of genetic ancestry with high altitude in the regulation of the vitamin D system, since vitamin D levels and placental metabolism were similar between the two genetic ancestry groups at sea level but different at high altitude. This suggests differential genetic adaptations to high-altitude stressors such as hypoxia and cold. Large cross-sectional studies in China [48] and Ecuador [28] revealed that Tibetans and Ecuadorian indigenous ethnicities showed the lowest 25-OH-D levels among all ethnicities studied. However, the interaction of altitude with ethnicity and alterations of vitamin D metabolites like 1α,25-(OH)_2_-D were not addressed in either study. We found vitamin D metabolism differences according to genetic ancestry in the ratio of 1α,25-(OH)_2_-D to 25-OH-D, in correlations between vitamin D metabolites, and in placental vitamin D-related gene expression. The 1α,25-(OH)_2_-D to 25-OH-D ratio was lower in both mother and fetus of European ancestry compared to Andean ancestry at high altitude. Recent studies have shown a stronger association between a low ratio of 1α,25-(OH)_2_-D:25-OH-D than just low 25-OH-D levels with cardiometabolic disease [49,50] and pregnancy disorders such as preeclampsia, intrauterine growth restriction, gestational diabetes, and preterm birth [41,51]. Our linear regression assays demonstrated that Andeans at high altitude had the highest correlation between 25-OH-D and 1α,25-(OH)_2_-D levels (both maternal and fetal) followed by the Andean-ancestry and European-ancestry groups (fetal only) residing at low altitude. In contrast, the European-ancestry group at high altitude showed the least correlation between the two metabolites suggesting higher variability and disturbed homeostasis in vitamin D metabolism within this group. Altogether, European-ancestry pregnancies are associated with adaptive changes in vitamin D metabolism more often associated with pregnancy complications and cardiovascular disease.

The differences in vitamin D metabolism between genetic ancestry groups at high altitude can be partly explained by the differential placental regulation of *CYP24A1*. This mitochondrial enzyme inactivates both 25-OH-D and 1α,25-(OH)_2_-D and was upregulated in placentas of European ancestry, but not those of Andean ancestry, at high altitude. Upregulation of *CYP24A1* has been observed in diseases where hypoxia is a key pathogenic component (e.g., cancer). Several studies have identified placental *CYP24A1* upregulation in preeclampsia, gestational diabetes, and fetal growth restriction in women residing at low altitude [52,53,54]. In healthy mammalian pregnancy, the increase in maternal 1α,25-(OH)_2_-D becomes uncoupled from *VDR*-mediated upregulation of *CYP24A1* in maternal renal and placental tissues, thereby preventing the inactivation of vitamin D metabolites [55,56,57]. Potential mechanisms of the differential effects of high altitude in Andeans and Europeans on the placental *VDR*/*CYP24A1* axis include crosstalk with pregnancy-specific hormones such as estrogen. Indeed, a recent study showed that, although high altitude increased estrogen and cortisol levels in near-term pregnant women, Andean women had higher estrogen and lower cortisol levels than European women [58]. Estrogen has been shown to correlate positively with vitamin D levels in human cohorts [59,60] and to upregulate *VDR* in neuronal tissues in vivo [61]. Therefore, future studies should identify the role of estrogen metabolites and other pregnancy hormones on differential vitamin D metabolism at high altitude according to genetic ancestry. Altogether, we have identified differential vitamin D metabolism in Andeans compared to European Bolivians in response to high-altitude stress, where the placenta plays a pivotal role. Because multiple pregnancy disorders with decreased vitamin D showed placental *CYP24A1* upregulation, future studies on placental *CYP24A1* regulation by hypoxia/high altitude are warranted. Previous metabolic studies have led us to hypothesize that high-altitude hypoxia alters placental utilization of oxygen in the mitochondria [62]. Since many vitamin D metabolizing enzymes are localized in the mitochondria, placental programming of mitochondrial activity/function could be partly responsible for the alterations in vitamin D related CYP450 expression, subcellular localization, and function.

Strengths of this study include the analysis of the vitamin D system in maternal/fetal dyads of different genetic-ancestry and altitude residences. Another important strength was the evaluation of vitamin D levels in correlation with placental mRNA expression of vitamin D-relevant genes that allowed us to uncover novel biological determinants of vitamin D homeostasis at high altitude, and that differed by genetic ancestry. The present study also has important limitations, for instance, skin color, dietary habits, and vitamin D supplementation, which usually have a strong effect on vitamin D status, were not measured. A second important limitation is that the small sample size prevented analysis of known gene polymorphisms in *VDR*, *CYP2R1*, *CYP24A1*, and other vitamin D-relevant genes in association with vitamin D levels. Multiple studies have revealed an association between these polymorphisms and vitamin D levels and disease [63,64,65]. Therefore, larger studies, including genetic polymorphisms are needed to validate the current data. A third limitation is the evaluation of vitamin D metabolites by ELISA instead of using the gold standard LC/MS. We have previously validated our commercial ELISAs using LC/MS in a subset of our samples and found that 1α,25-(OH)_2_-D levels are overestimated by ELISA as compared to LC/MS. However, the differences between groups remained the same [55,57]. Finally, all of our subjects were healthy-term pregnant women and their singleton neonates. It would be of great interest to investigate the effect of altitude on vitamin D homeostasis and metabolism in pregnancies complicated with fetal growth restriction or other pregnancy disorders.

## 4. Materials and Methods

### 4.1. Study Design

This study involved 40 maternal/fetal dyads that were part of a larger, cross-sectional study evaluating the effects of altitude and genetic ancestry on oxygen delivery/consumption, uterine/umbilical blood flows, placental efficiency, and pregnancy outcomes [39]. Genetic ancestry was determined with 133 single nucleotide polymorphisms as previously described [39]. This study was approved by the Bolivian National Bioethics Committee, and local and the US Institutional Review boards, and all the participants gave written and informed consent.

All study participants were screened for chronic health conditions and general health 0–10 days prior to scheduled Cesarean delivery [37]. Women were excluded for drug, alcohol, or tobacco use, complications of pregnancy, a positive oral GTT test, pre-term labor, or premature rupture of the membranes. The European group was composed of women of sea level ancestry with no known Andean ancestors and living at ≤400 m for three generations, similar women who migrated to 3600 m, and whose Ancestry-Informative Markers (AIMS, [37]) indicated predominantly European ancestry. The Andean group was comprised of a similar group. Women at high altitude, with three generations of ancestors born and raised at high altitude who either remained in their native environment or migrated to ≤400 m as children or adults, and whose ancestry was predominantly Native American. Maternal blood was collected from the antecubital vein after consent was obtained, during the screening visit 4 ± 1 day prior to delivery (scheduled Cesarean). Umbilical cord blood (fetal/neonatal) was collected from the doubly-clamped umbilical cord vein immediately after delivery. Blood samples were placed in heparinized tubes and processed to collect plasma aliquots that were frozen at −70 °C until further analysis. Placental samples were also obtained from four locations (one from each of the quadrants bisected by the cord insertion), snap-frozen in liquid nitrogen, and stored at −70 °C until further analysis.

### 4.2. Vitamin D Status and Vitamin D Binding Protein ELISA Assays

Plasma levels of 25-OH-D and 1α,25-(OH)_2_-D were analyzed using commercially available EIA kits that determine total (bound and free) levels of vitamin D metabolites (Immunodiagnostic Systems, Ltd., Scottsdale, Arizona). The sensitivity of the 25-OH-D ELISA kit is 5 nM, the intra- and inter-assay variability is ≤8.7% coefficient of variation (CV), and the specificity is 100% for 25-OH-D_3_, 75% for 25-OH-D_2_, 100% for 24,25-(OH)_2_-D, and less than 0.3% for the remaining vitamin D metabolites. The sensitivity of the 1α,25-(OH)_2_-D ELISA kit is 6 pM with a specificity of 100% for 1α,25-(OH)_2_-D_3_, 39% for 1α,25-(OH)_2_-D_2_, less than 0.05% for the remaining vitamin D metabolites, and the intra- and inter-assay variability is estimated at ≤20% CV.

Vitamin D-binding protein (VDBP) was determined with a commercial ELISA kit (Aviva Systems Biology LLC, San Diego, CA, USA) using the manufacturer’s instructions.

### 4.3. Placental Vitamin D-Related Gene Expression Analysis

Placental biopsies were ground to obtain a homogenous representative sample. RNA isolation, RT reaction, and real-time PCR were performed as previously described [41]. Samples were analyzed on the CFX Connect™ system (Bio-Rad, Hercules, CA, USA) using the QuantiTect^®^ Probe PCR kit (Qiagen, Hilden, Germany), and Taqman primer sets (Thermo Fisher Scientific, Waltham, MA, USA) according to the manufacturer’s protocol. Negative controls were included in each run. Gene expression was determined for *DHCR7* (Hs01023087_m1), *CYP2R1* (Hs01379776_m1), *LRP2* (Hs00189742_m1), *CUBN* (Hs00153607_m1), *GC* (Hs00167096_m1), *CYP27B1* (Hs01096154_m1), *CYP24A1* (Hs00167999_m1), and *VDR* (Hs01045840_m1). The housekeeping gene β-actin (*ACTB*, Hs01060665_g1) was used to normalize target mRNA levels. Quantitative analysis was performed with the aid of standard curves as previously described [41].

### 4.4. Statistical Analysis

Quantitative data are shown as mean and standard error. One-way ANOVA with LSD posthoc analysis was used to compare the four groups on quantitative characteristics of mother and baby, vitamin D status, and mRNA/protein expression variables. Non-parametric data were analyzed with Kruskall–Wallis One-way ANOVA. Two-way ANOVA was used to determine the significance of ethnicity, altitude, and the interaction between them. Chi-square was used for comparisons of categorical characteristics (i.e., fetal sex). Simple linear regression models were developed to examine the relationships between maternal and umbilical cord blood vitamin D levels, and between stable precursor (25-OH-D) and most active (1α,25-(OH)_2_-D) vitamin D metabolites for each of the four groups. To determine potential predictors of vitamin D levels, multivariable linear regression analysis was performed, and the model with the highest adjusted r^2^ while showing significance (*p* < 0.05) was chosen. IBM SPSS statistics version 26 was used.

## Figures and Tables

**Figure 1 ijms-24-03389-f001:**
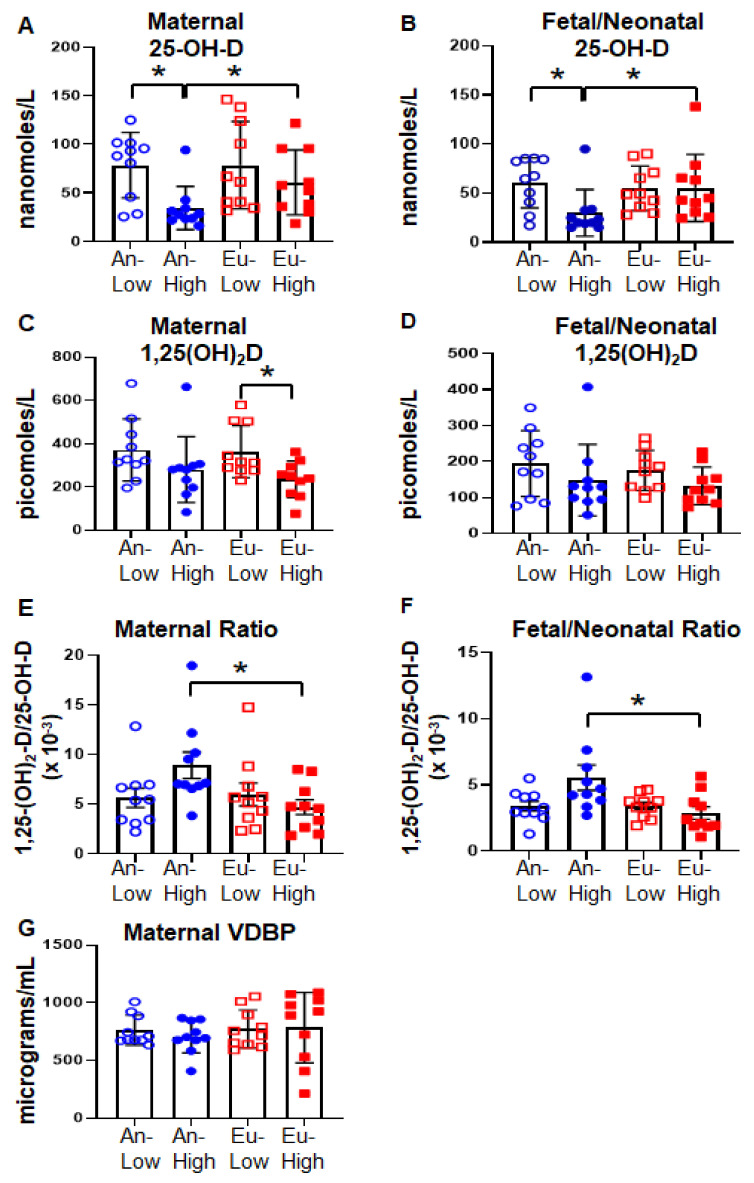
Vitamin D status in Bolivian pregnancies. Vitamin D metabolite circulating levels were determined in maternal and umbilical cord blood of 40 mother/fetus dyads divided into four groups according to altitude residence and genetic ancestry via ELISA as explained under methods. (**A**) maternal 25-OH-D, (**B**) umbilical cord 25-OH-D, (**C**) maternal 1α,25-(OH)_2_-D, (**D**) umbilical cord 1α,25-(OH)_2_-D, (**E**) maternal ratio of 1α,25-(OH)_2_-D/25-OH-D, (**F**) umbilical cord ratio of 1α,25-(OH)_2_-D/25-OH-D, and (**G**) maternal VDBP levels. Bars represent the average ± standard error (*n* = 10/group). * *p* < 0.05 as determined by one-way ANOVA followed by LSD post-hoc analysis.

**Figure 2 ijms-24-03389-f002:**
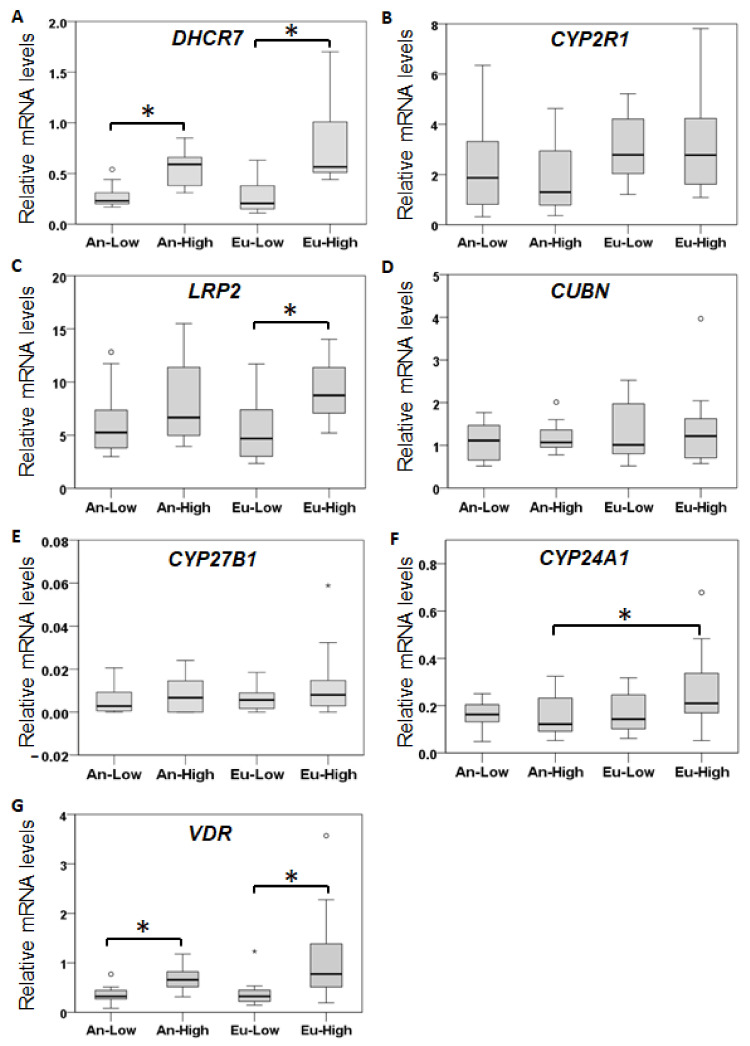
Placental expression of vitamin D-related genes. Placental mRNA levels, shown relative to *ACTB* mRNA levels (× 10^−3^), were determined by Taqman probe qPCR as described under methods. Data is shown in boxplots for *DHCR7* (**A**), *CYP2R1* (**B**) *LRP2* (**C**), *CUBN* (**D**), *CYP27B1* (**E**), *CYP24A1* (**F**), and *VDR* (**G**). * *p* < 0.05 as determined by one-way ANOVA followed by LSD post-hoc analysis.

**Figure 3 ijms-24-03389-f003:**
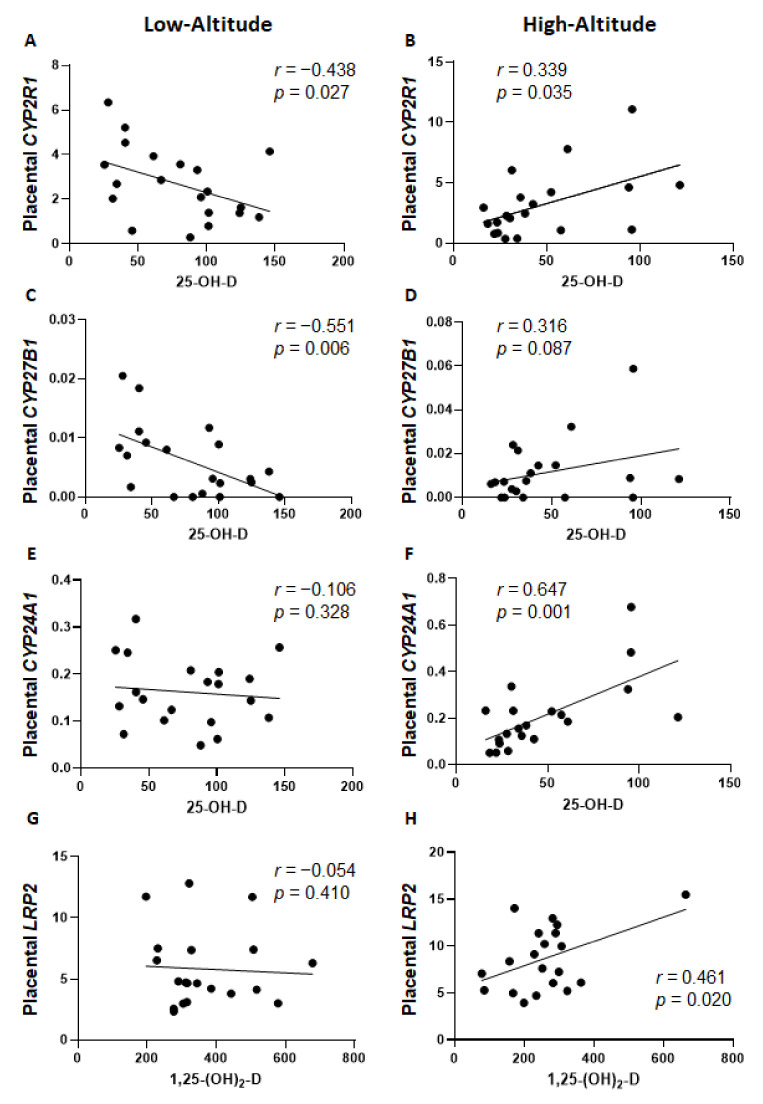
Correlation between vitamin D levels and placental gene expression. Linear regression analysis shows the association in low-altitude residing subjects (*n* = 20, (**A**–**D**)) or high-altitude residing subjects (*n* = 20, (**E**–**H**)). (**A**,**B**) correlation between maternal 25-OH-D and placental *CYP2R1*, (**C**,**D**) maternal 25-OH-D and *CYP27B1*, (**E**,**F**) maternal 25-OH-D and *CYP24A1*, and (**G**,**H**) maternal 1α,25-(OH)_2_-D and *LRP2*. Insets show the correlation coefficient and *p*-value.

**Table 1 ijms-24-03389-t001:** Maternal and Fetal/Neonatal Characteristics of the Study Groups.

Characteristics ^a^	Andean Ancestry at 400 m Altitude (*n* = 10)	European Ancestry at 400 m Altitude (*n* = 10)	Andean Ancestry at 3600 m Altitude (*n* = 10)	European Ancestry at 3600 m Altitude (*n* = 10)	*p*-Value
Maternal					
Age (yr)	29.1 ± 1.7	25.6 ± 1.7	32.4 ± 1.6	30.0 ± 1.7	0.008 *
Pre-pregnancy BMI (kg/m^2^)	23.1 ± 0.8	23.3 ± 1.0	26.4 ± 2.1	23.4 ± 0.9	0.277 *
Fetal/Neonatal					
Sex (% Male)	60	70	50	50	0.771 ^†^
Gestational age (wk)	39.5 ± 0.4	38.4 ± 0.3	39.1 ± 0.4	37.9± 0.3	0.016 *
Birthweight-BW (g)	3491 ± 106	3453 ± 94	3446 ± 119	2939 ± 95	0.001 *
BW Percentile (%)	52.1 ± 7.0	63.6 ± 8.6	51.4 ± 7.6	31.0 ± 6.2	0.030 *
BW:PW ratio	7.54 ± 0.41	7.76 ± 0.28	6.42 ± 0.38	6.23 ± 0.46	0.016 *
Ponderal Index	2.65 ± 0.09	2.73 ± 0.05	2.84 ± 0.10	2.48 ± 0.06	0.022 *
Head Circumference	35.1 ± 0.5	34.7 ± 0.3	34.6 ± 0.3	34.0 ± 0.5	0.387 *
Ab. Circumference	34.7 ± 0.5	33.9 ± 0.3	34.9 ± 0.2	33.1 ± 0.4	0.005 *

^a^ Characteristics with continuous data are shown as mean ± standard error. BW = birthweight; PW = placental weight; Ab = abdominal; * One-way ANOVA; ^†^ Chi-square test.

**Table 2 ijms-24-03389-t002:** Demographic and clinical determinants of vitamin D levels in Bolivian pregnancies.

Dependent Variable (*n* = 40)	Model * Adjusted R^2^	Beta Coefficient	Confidence Interval 95%	*p*
Maternal 25-OH-D	0.149			0.008
1. Altitude		30.9	8.6–53.3	0.008
Fetal/neonatal 25-OH-D	0.156			0.016
1. Altitude		26.4	7.1–45.7	0.009
2. BW: PW		−8.5	−15.6–(−1.4)	0.020
Maternal 1α,25-(OH)_2_-D	0.282			<0.001
1. Altitude		168	83.8–252.5	<0.001
2. BW: PW		−44.0	−75.1–(−12.8)	0.007
Fetal/neonatal 1α,25-(OH)_2_-D	0.256			0.002
1. Altitude		84.9	35–134.7	0.001
2. BW: PW		−30.3	−48.7–(−11.9)	0.002

* Multivariable linear regression was performed to determine potential factors of vitamin D levels, and the model with the highest adjusted R^2^ was chosen. The following factors were studied: altitude, genetic ancestry, maternal age, maternal pre-pregnancy BMI, fetal sex, gestational age, birthweight, birthweight percentile, birthweight-to-placental weight ratio (BW: PW), and ponderal index.

**Table 3 ijms-24-03389-t003:** Placental determinants of vitamin D levels in Bolivian pregnancies.

	Model * Adjusted R^2^	Beta Coefficient	Confidence Interval 95%	*p*
Low Altitude Subjects (*n* = 20)				
Maternal 25-OH-D	0.265			0.012
1. *CYP27B1*		−3527	−6173–(−881)	0.012
Fetal/neonatal 25-OH-D	0.221			0.021
1. *CYP2R1*		−7.52	−13.79–(−1.26)	0.021
High-Altitude Subjects (*n* = 20)				
Maternal 25-OH-D	0.505			0.001
1. *CYP24A1*		118.7	50.0–187.4	0.002
2. *CYP2R1*		6.18	0.54–11.8	0.034
Fetal/Neonatal 25-OH-D	0.351			0.010
1. *CYP2R1*		7.75	1.14–14.4	0.024
2. *CYP24A1*		80.7	0.1–161.3	0.050
Maternal 1α,25-(OH)_2_-D	0.297			0.019
1. *LRP2*		15.98	1.49–30.47	0.033
2. *CYP24A1*		314.6	−6.1–635.2	0.054

* Multivariable linear regression was performed to determine potential placental gene determinants of vitamin D levels, and the model with the highest adjusted R^2^ was chosen.

## Data Availability

Not applicable.

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
