# Peer review of "Decreased Vitamin D Levels and Altered Placental Vitamin D Gene Expression at High Altitude: Role of Genetic Ancestry"

_ijms, 2023, doi:10.3390/ijms24043389_

Round 1
Reviewer 1 Report
In this manuscript, the authors investigate the effect of altitude and genetic ancestry on maternal/fetal vitamin D levels and placental vitamin D metabolism in Andeans and Europeans women. They confirmed previous reports that high-altitude residence is associated with decreased vitamin D levels in both mother and fetus with Andean-ancestry subjects having decreased 25-OH-D and European-ancestry subjects decreased 1α,25-(OH)2-D.
Moreover, they identified placental efficiency as a secondary determining factor fetal 25-OH-D and maternal/fetal 1α,25-(OH)2-D levels, which led them to evaluate placental gene expression of the vitamin D system and conclude that the placenta has an important role in regulating circulating levels of vitamin D metabolites in both mother and fetus.
This study is a small piece of the puzzle; however, it may contribute to our understanding of the association and potential mechanism of high-altitude effects on maternal and fetal/neonatal vitamin D status and placental vitamin D metabolism however, future studies are needed.
The paper is generally well written and structured. However, before the publication the article small comments are needed:
- The authors consider the possible cross-talk between vitamin D and estrogen. In the study, the authors analyze the levels of estrogen and/or vitamin D-related genes (CYP24A1, VDR, CYP27B1) in pregnant women, but only at the placental level. It is very important, because studies in humans and in animal models showed that 17β-estradiol is able to decrease the expression of CYP24A1 and increases the expression of VDR gene in diverse human and rat tissues. The relationship between 17β-estradiol and vitamin D is further supported by (i) the significant increase of 25(OH)D levels observed in women assuming estrogen containing contraceptives [Harmon QE et al., 2016], and (ii) the association between low 25(OH)D levels and low 17β-estradiol levels in women [Zhao D, et al., 2017]. On the other hand, 1,25(OH)2D affects in a tissue-specific way peripheral estrogen metabolism [Lundqvist J et al., 2011]. In mouse splenic T lymphocytes, 1,25(OH)2D enhances the expression of the CYP19 gene encoding aromatase, the enzyme that produces 17β-estradiol from testosterone [Spanier JA et al., 2015]. In addition, Charles SM et al. (Charles SM et al. 2014), in their study show that high-altitude residence increase circulating progesterone, cortisol, estrone, 17β-estradiol, and estriol levels as the result of greater aromatase activity.
For this reason it would be interesting to study the serum levels of sex hormones in pregnant women, considering that estrogen levels are particularly high in pregnant women, the reduction in vitamin D levels observed at high altitude could in part be explained by this correlation between vitamin D and estrogen.
Author Response
We thank this reviewer for the insightful comments on vitamin D metabolism, in particular the potential role of estrogen in mediating the effects of high altitude on vitamin D metabolism in Andeans compared to European-Hispanics.
We have made additions to the discussion in lines 248-254 as shown below, and hope that is a satisfactory review of this manuscript.
'Potential mechanisms of the differential effects of high altitude in Andeans and Europeans on placental VDR/CYP24A1 axis include crosstalk with pregnancy-specific hormones such as estrogen. Indeed, a recent study showed that although high altitude increased estrogen and cortisol levels in near-term pregnant women, Andean women had higher estrogen and lower cortisol levels than European women [58]. Estrogen has been shown to correlate positively with vitamin D levels in human cohorts [59,60] and to upregulate VDR in neuronal tissues in vivo [61]. Therefore, future studies should identify the role of estrogen metabolites and other pregnancy hormones on differential vitamin D metabolism at high altitude according to genetic ancestry.'
We included 3 of the references suggested by the reviewer and included one additional on estrogen effects on VDR. We could not find references on estrogen downregulating CYP24A1, but instead found the opposite effect of upregulation. Nevertheless, we agree with the reviewer and future studies should consider the effects of high altitude on vitamin D via hormonal cross-talk.

Reviewer 2 Report
In this manuscript, the authors aimed to investigate the effect of altitude and genetic ancestry on maternal/fetal vitamin D levels and placental vitamin D metabolism. They demonstrated differential vitamin D metabolism in Andeans compared to European Bolivians in response to high altitude.
The study is well designed and results are straightforward and clearly presented. Overall, the manuscript is well written. I have no further comments.
Author Response
We thank this reviewer for the kind words and appreciation of this research study.
We attach the most current version of this study with additional discussion requested by the other reviewer, shown in lines 248-254.
